# Language distance and labor market integration of migrants: Gendered perspective

**Eyal Bar-Haim** [1], **Debora Pricila Birgier** [2] *

1 Arlozorov Forum, Faculty of Educational, Bar-Ilan University, Ramat Gan, Israel, 2 Department of Sociology, Demography Unit (SUDA), Stockholm University, Stockholm, Sweden

* debora.birgier@gu.se

**Data Availability Statement:** All files are available from the www.oecd.org database using the following URL: https://doi.org/10.4232/1.12955.

**Funding:** We gratefully acknowledge the financial support provided by the Israel Science Foundation

## Abstract

This paper examines the distinct effects of linguistics distance and language literacy on the labor market integration of migrant men and women. Using data from the Programme for International Assessment of Adult Competencies (PIAAC) 2018 in 16 countries of destination mainly from Europe and more than 110 languages of origin, we assess migrant labor force participation, employment, working hours, and occupational prestige. The study finds that linguistics distance of the first language studied has a significant negative association with labor force participation, employment, and working hours of migrant women, even after controlling for their abilities in their destination language, education, and cultural distance between the country of origin and destination. In contrast, linguistics distance is only negatively associated with migrant men's working hours. This suggests that linguistic distance serves as a proxy for cultural aspects, which are not captured by cultural distance and hence shape the labor market integration of migrant women due to cultural factors rather than human capital. We suggest that the gender aspect of the effect of language proximity is essential in understanding the intersectional position of migrant women in the labor force.

## 1. Introduction

This paper aims to understand the importance of language distance for the labor market integration of immigrants. Language distance between origin and destination was found to be associated with overall migration flows [1–3], migrants' language acquisition at the destination [1, 4–6], social integration [7], and labor market outcome [7, 8]. Along the same line, language abilities and literacy are among the most critical aspects of migrants' integration at their destination, and several migration studies show that language ability and literacy substantially affect migrants' labor market performance [9]. While some studies indicate that the effect of linguistics distance on labor market outcome is a result of lower host country language acquisition of migrants [8, 10, 11], others focus on linguistics distance as a proxy for cultural distance [14–16, 22–24]. Thus, language is usually an overlooked form of cultural capital. Moreover, studies examining heritage language use in the context of the gender-immigration nexus argue that it

(Grant No. 80/20) and the Swedish Research Council for Health, Working Life, and Welfare (FORTE) (Grant No. 2016-07105) for this work. The funders played no part in the study design, data collection and analysis, decision to publish, or preparation of the manuscript.

**Competing interests:** The authors have declared that no competing interests exist.

is associated with gender norms that shape migrant women's integration into the labor market [12–15]. This might indicate that literacy captures a fraction of migrants' social assimilation, shaping migrants' economic integration. While linguistics distance captures additional aspects, which might be associated with an accent, orientation, and norms, potentially affecting labor market outcomes, such as labor force participation, employment, working hours, and occupational prestige. In this paper, we ask whether linguistics distance has a distinct effect from the host country's language proficiency on migrants' labor market assimilation due to its role as a proxy for cultural distance and cultural capital. We pay specific attention to gender differences in the relation between linguistics distance and labor market outcome due to the unique position of migrant women.

## 2. Theoretical background

### 2.1 Language distance and labor market outcome: Human or cultural capital?

Language distance can shape migrants' integration at their destination in three primary ways. *First*, it may indirectly impact migrant economic integration through its influence on language acquisition. Many studies have found that greater linguistic distance is associated with larger disparities in language proficiency and often the slower acquisition of the destination language [5, 6]. These findings have been consistently observed in studies using a single-country approach [4, 5, 16], a multiple origin-multiple destination design in a double comparative approach [6], and alternative measures of linguistic distance. This supports the notion that learning languages that are linguistically distant from one's mother tongue is more challenging. The association between language proficiency and immigrants' labor market outcomes has been widely studied in many countries, mostly indicating a direct causal effect on earnings, with the size of the effect ranging from 5 to 30 percent (for an overview of empirical findings, see [9]).

*Second*, language distance can directly impact the economic integration of migrants in their destination country. Individuals with greater language distance may find it difficult to obtain employment and have better occupations and higher wages, as the transferability of human capital is more accessible when the linguistic gap is smaller [8, 10, 11]. Surprisingly, proximity to English was not found to have a consequence on economic integration, stressing the importance of being fluent in the local language [8]. Additionally, migrants may choose occupations where their language barrier is less influential for their success [7]. Interestingly, the effect of language distance on migrant integration is evident even in the long term and for childhood immigrants who are expected to have time to learn the native language. For example, it was found that linguistic distance interacts with age at arrival to shape the occupational outcomes and choice of college major of childhood immigrants from different countries [17].

*Lastly*, some studies perceive language distance as a source of discrimination [18]. According to this tradition of studies, the linguistic distance between the immigrant and the host country's language serves as a cultural signal that enables employers to discriminate against the immigrant even if his or her host country's language proficiency is high [19]. Hence, language distance or proximity should be regarded as a form of cultural capital or linguistic capital. Cultural capital is a term dating back to Bourdieu's [20] work on educational inequalities. It represents the ability to signal traits of the dominant culture by a student (in our case—the employee) in a way that would be received positively by the authority—the teacher or the employer, who is a part of the dominant culture. In that regard, we can see both language ability and language distance as measuring two distinct aspects of cultural capital. In line with this argument Schmaus (2020) investigated the differential impact of language skills on labor

market success among various groups of migrants, considering variations in their level of associated distaste by employers. They suggest language proficiency might also be linked to taste discrimination against specific ethnic groups [21]. The control for cultural distance enables us to study the direct effect of the performative aspect of culture as it is reflected in the language itself.

All three perspectives suggest that language distance directly or indirectly affects immigrants' labor market outcomes. Unfortunately, most studies do not empirically control for language ability to assess the clean effect of linguistic distance on labor market outcomes, nor do they try to assess if the effect is associated with cultural distance. In addition, several questions remain unanswered. For instance, does language distance still affect the labor market outcome of migrants once their language proficiency and cultural distance are taken into account? Does language distance have different impacts on various labor market outcomes, such as labor market participation, employment, working hours, and occupational prestige? Additionally, what are the effects on the labor market outcomes of migrant men and women?

We hypothesize that language distance may not only impact labor market outcomes through facilitating language acquisition but also through its association with cultural capital and have a distinct effect by gender. Hence, this study aims to investigate the impact of linguistic distance by looking at the first language studied at home, on the labor force status of migrants stratified by gender. Specifically, we explore how linguistic distance, independent of literacy skills in the destination country language and cultural distance between source and destination, influences migrants' labor force participation, working hours, and occupational prestige. In the following section, we will delve into the potential gender variations regarding the correlation between linguistic distance and the labor market integration of migrants.

## 2.2 Culture and migrant women labor market integration

Migration and feminist scholars have extensively studied the unique experiences of migrant women in the association of gender and migration using different terms. The first is "double disadvantage," which refers to labor market disadvantages migrant women have compared to both male migrants and native women. It was suggested that since migrant families tend to invest more in the husbands' labor force assimilation, married migrant women, especially with children, are more prone to suffer from double disadvantage [22–24]. The second term is "intersectionality," which refers to the unique experience of disadvantaged subgroups (for example, women) within a minority or disadvantaged group. Intersectionality, as a concept extending beyond gender and migration, serves as a crucial lens for understanding the intricate web of challenges individuals face in the labor market. While intersectionality is not restricted to migrant women but rather unfolds when as factors like race, age, and qualifications intersect to shape the experience of individuals, in this paper we focus on the interaction between migration and gender and how it might be shaped by linguistic distance. Following this tradition, immigrant women face different barriers but also opportunities than native women and immigrant men [25, 26]. For instance, the convergence of gender-related discrimination and language barriers might significantly impact the journey of migrant women as they strive to integrate into the labor market. To illustrate, women hailing from specific cultural backgrounds may confront gender-specific biases that intertwine with linguistic differences, thereby amplifying the complexity of their employment endeavors. Importantly, both traditions call for examining the experience of migrant women in light of gender perceptions and family roles.

There are two primary mechanisms by which language distance might shape the integration of migrant women (somewhat different than men) in the labor market. From what we term

the *cultural capital perspective*, language difference is seen as a form of cultural advantage. The ability to pass as a native, or to come from a similar background as locals, becomes the basis for discrimination in the labor market [19, 27]. While the cultural capital perspective is relevant both to men and women, we believe that the implications are more substantial for migrant women [28]. On the other hand, scholars who adopt what we call the *gender cultural norms* approach perceive language distance as a measure of cultural characteristics that are important to gender division in the labor market. These scholars mainly highlight the cultural trait of family-work division, which might be reflected in language distance [12, 15].

While we suggest that cultural capital might be the mechanism by which language distance might shape both migrant men's and women's labor market integration, previous studies suggest that cultural capital is more important for women than for men [28]. We propose that the performative effect of language, or the perception of the host country's language as cultural capital, is expected to affect women more than men for several reasons. First, studies have shown that women, and immigrant women are no exception in this regard, tend to concentrate in occupations where communication skills are more important, for example, in the service industry than their male counterparts [29–31]. This implies that immigrant women are more prone to be discriminated against in the labor market due to language distance since their position in the labor market is highly dependent on communication skills [32]. This kind of discrimination, especially in occupations that require intensive communication skills which are traditionally feminine, was found in various countries [18, 27, 33]. For instance, the role of language in the discrimination of migrant women was demonstrated in Australia, where Dovchin (2019) described how Mongolian women, some of them with high proficiency in English, experienced racism and discrimination due to their heavy accents, which perceived as "broken English" [27].

Second, migrant women encountered more significant language barriers to their participation in the labor force, particularly in terms of speaking and comprehension skills [7]. Discrimination related to language use against immigrant women exists for both high and low-skilled workers, albeit in different forms. In Canada, for example, Man (2004) describes a process of "deskilling" of immigrant women of Chinese background with high skills. This is done by various institutionalized processes, such as a demand for "Canadian experience" for eligibility to feminine occupations [19]. Similarly, a recent study finds that limited proficiency in the Italian language had a more detrimental effect on immigrant women's labor market outcomes than immigrant men [7].

The *gender cultural norms perspective* examines how migrant women's labor market outcomes in their destination are shaped by gender norms from their source country. Numerous studies, primarily in the US, have explored how differences in female labor force participation rates across source countries contribute to disparities in the labor market behavior of immigrant women at their destination. These studies underscore that variations arise from cultural perceptions about women's roles, influencing the labor market behavior of immigrant women and their descendants at their destination [34–37].

Most of these studies majored cross-country variations in cultural beliefs regarding women's roles by using women's labor force participation in their source countries. For example, Blau and Kahn (2015) [38] use female-to-male LFP ratios as a cultural proxy to investigate the effect of human capital and culture on the labor supply and wages of immigrant women in the US. They found that women from source countries with higher FLFP have higher working hours in the US, and this effect remains after controlling for the immigrant's own pre-migration labor supply. In addition, it was found that the effect of source country culture trickles down to second and higher-generation and persists in the long run [34, 39, 40]. While most of these studies have been done in the US framework, recently, a few studies have addressed this

question in Europe [35, 41, 42]. Bredtmann and Otten, (2023) explore the same question in different European countries and found a positive correlation between the female-to-male labor force participation ratio in the source country and migrant women's labor supply, which does not persist through the second generation [43].

Migrant families might maintain their origin cultures in several ways, and speaking their heritage language is one way to do so [44]. A heritage language is not only a means for the intergenerational preservation of culture but also an indicator of cultural assimilation [15]. Recent studies suggest that heritage language can be used as an indicator of cultural traits related to the division of work in the family [12–15]. It was found that second-generation migrant women who use their heritage language at home were less prone to participate in the labor market and work fewer hours [15]. Along the same lines, speaking a language with gender-based grammatical roles was associated with lower labor market participation and working hours of migrant women [12, 13]. Therefore, we use language distance as a proxy for cultural norms, including gender norms. In that line of argument, controlling for other aspects of cultural distance enables us to assess the distinct effect of linguistic distance. While our focus is not on language used at home, we believe examining first language acquisition at home might capture childhood exposure to gender norms.

Both the *cultural capital* and *gender cultural norms* perspectives predict that immigrant women will have higher language-related disadvantages in the labor market due to linguistic distance. Moreover, these perspectives also predict that the effect of language distance on immigrant women's performance in the labor market will be net of linguistic proficiency in the host country's language and cultural distance between source and destination. Essentially, the critical distinction between these approaches lies in the role of agency: while the former scholars place greater emphasis on labor market discrimination and the employers' tendency to prefer native language speakers (e.g., focusing on the demand side of the labor market), the latter emphasizes the agency of immigrants and their cultural preferences (e.g., focusing on the supply side of the labor market). As such, linguistic distance effect lies in the interaction between the supply and demand, depending on the perspective in which we focus on.

## 3. Comparison strategy and expectations

The literature leads to the following hypotheses regarding the association between linguistic distance, linguistic ability, and labor market outcome by gender:

$H_1$: Higher linguistics distance will be associated with lower levels of LFP, employment, working hours, and occupational prestige of migrants controlling for their actual language abilities.

$H_2$: Migrant women will have lower levels of LFP, employment, working hours, and occupational prestige when the linguistic distance is larger relative to migrant men due to the association of cultural distance and gender norms.

$H_3$: If the *cultural capital perspective* serves as the primary mechanism influencing the integration of migrant women, it emphasizes a demand-driven explanation that includes labor market discrimination affecting their entry into the workforce. In that case, we anticipate observing the impact of linguistic distance on various aspects of migrant women's labor market integration, including labor force participation (LFP), employment, and occupational prestige. To a lesser degree, we expect linguistic distance to influence working hours, reflecting the role of labor market discrimination in the initial entry of migrants into the workforce, with a comparatively diminished impact on their working hours post-employment.

H₄: Alternatively, if *gender cultural norms* play a pivotal role in shaping the integration of migrant women, they will particularly influence their labor force participation (LFP), employment status, and working hours—representing supply-driven factors—and we anticipate observing the impact of linguistic distance on these aspects. Occupational prestige is expected to be influenced to a somewhat lesser extent, reflecting the demand side of the labor market, while the first three outcomes primarily align with labor supply decisions as women decide their involvement in the labor market.

## 4. Data, variables, and methods

### 4.1 Data and sample

In order to test these expectations, we use the Programme for International Assessment of Adult Competencies (PIAAC) 2018 which contains information from 36 countries and territories. We restricted the sample to immigrants at their prime working age, resulting in 4,843 observations in 16 countries of destination coming from more than 110 languages of origin, from which we have information on linguistics distance and sufficient numbers of migrants. The benefit of using the PIAAC data set relative to alternative data is that the PIAAC data contain an assessment of actual linguistic literacy. In addition, individuals in the PIAAC data were asked about their mother tongue and could name up to two options. The question worded as follows: "What is the language that you first learned at home in childhood and still understand?"

We used this information as the basis for matching the linguistic distance. It is imperative to acknowledge that the assessment of linguistic proficiency derived from PIAAC is contingent upon the language of the destination. Consequently, the consideration of endogeneity issues becomes pivotal, given the sample's constraint to individuals possessing the requisite proficiency to undertake the evaluation (e.g., those with sufficient linguistic competence to comprehend the posed questions). In addition, a study conducted in Germany suggests that the response rate for the PIAAC of migrants is lower than that of natives [45].

### 4.2 Variables

In order to obtain the language distance variable, we applied the dataset created by Melitz and Toubal for language proximity [46]. The dataset is a matrix that contains information on the common language spoken in each country and its linguistic proximity with every other country, calculated using ASJP scoring of similarity Bakker et al. [47]. This method compares a list of between 100 to 200 words in two languages to identify cognate words and calculates the percentage of similar words (see: [48]). The linguistics distance scale ranges from zero to one, with a larger value representing greater linguistics distance. Using data obtained from the Alveo Virtual Laboratory [49], which matches languages to countries, we assigned each language in the PIAAC dataset to the relevant country and added the proximity score for each migrant based on their declared language learned at home resulting in an origin language by host country language score for each individual. For example, the smallest distance is between speaking Croatian in Slovenia (0.13), while the largest distance is between speaking Burmese in Norway (0.89) or Eritrean in the UK (0.88). Note that the most frequent language used in the country determines the host country's language. Cases where the respondent learned more than one language were treated by the first language the respondent learned and still speaks. In addition, to have a more balanced distribution of linguistics distance, cases in which individuals spoke the same language at the origin and the host country were omitted from the analysis.

To discern the influence of language distance and cultural distance, we incorporate control measures for cultural distance utilizing the World Value Survey. The Inglehart et al. (2014) [50] exes of cultural distance between countries are employed for this purpose. Given that certain countries possess data spanning multiple waves of the World Value Survey, we prioritize information from the 2008 year or the nearest available year. In instances where this specific year is unavailable and only one year is accessible, we utilize the available information. However, it is imperative to note two significant caveats associated with the cultural distance variable. Firstly, the World Value Survey does not encompass all source countries included in the PIAAC, leading to a reduction in the number of cases in models incorporating control for cultural distance. Secondly, for some migrants the absence of information on individual place of birth when employing control for cultural distance further narrows our sample. Finally note that while linguistic distance and cultural distance might be related, they are two separate aspects for several reasons. First, linguistic distance is based on the first language that the individual learned at home (aiming at capturing the mother tongue), and cultural distance is based on the country of birth of the individual. Second, individuals having the same first language might come from different countries and hens have different cultural distances, for example, two individuals living in Sweden whose language is Spanish but one of them was born in Spain and the other in Chile.

As our focus lies on examining the impact of linguistic distance on the measured literacy of the destination language, we incorporate various control variables. Firstly, we account for individual scores on the literacy test. Additionally, we consider the duration of migrants' stay at their destination (more than ten = 0, vs. less than ten years at destination), age, educational attainment, and whether their highest level of education was obtained abroad. Finally, as we are interested in aspects related to gender, we also controlled for living with a partner and having children in the household. Appendix 1 in S1 File provides a descriptive table of all the variables used in the analysis by gender.

## 4.3 Methods

To unravel the mechanisms underlying the relationship between linguistic distance and labor market outcomes among migrant men and women, our analysis was conducted in several stages. Initially, we examined the association between linguistic distance and labor force participation, employment, working hours, and occupational prestige for both male and female migrants. In these analyses, we place particular emphasis on gender differences regarding the impact of linguistic distance on these outcomes, controlling for language abilities. For labor force participation and employment outcomes, we employed linear probability models, while we utilized linear regression models for working hours and occupational prestige outcomes, we incorporated destination country-fixed effects in all models. We first run models for all individuals and next include the gender interaction with linguistic distance. Subsequently, we conducted separate analyses by gender. Finally, we add to the models by gender a control for cultural distance to assess whether the effect of linguistic distance remains significant after controlling for cultural distance.

## 5. Findings

The subsequent section provides a comprehensive overview of our findings. Initially, we examine the impact of linguistic distance on labor force participation and employment, trying to establish a significant relationship between language distance and the economic integration of migrants. Subsequently, we investigate the association between linguistic distance and working hours, an aspect documented in the literature to be more associated with individual preference

variables rather than a consequence of discrimination [15]. Finally, we present the outcomes of our analysis concerning occupational standing (ISEI), an indicator that, according to existing literature, might be more influenced by discriminatory practices directed towards migrants [51].

Table 1 presents the findings pertaining to labor force participation. As can be seen from Model 1, the language distance decreases the probability of participation in the labor market significantly, net of language proficiency and gender, as well as all the other socio-demographic characteristics. The effect of gender is significant, indicating that migrant women are less likely to participate in the labor market than migrant men, net of language distance.

Nevertheless, with the inclusion of an interaction term in the model (Model 2), the initial significance and strength of the main effect of language distance diminishes. Instead, the

**Table 1. Labor force participation of migrants by linguistics distance.**

| VARIABLES | (1)<br>All | (2)<br>All | (3)<br>Women | (4)<br>Men | (5)<br>Women | (6)<br>Men |
|---|---|---|---|---|---|---|
| Linguistics distance | -0.249*** | -0.025 | -0.568*** | 0.106* | -0.536*** | 0.124 |
| | (0.046) | (0.057) | (0.068) | (0.059) | (0.087) | (0.082) |
| Female | -0.181*** | 0.148*** | | | | |
| | (0.012) | (0.052) | | | | |
| Female *Linguistics distance | | -0.450*** | | | | |
| | | (0.069) | | | | |
| BA | 0.079*** | 0.076*** | 0.092*** | 0.028 | 0.100*** | 0.036 |
| | (0.018) | (0.018) | (0.025) | (0.024) | (0.027) | (0.027) |
| MA+ | 0.118*** | 0.118*** | 0.094*** | 0.142*** | 0.120*** | 0.156*** |
| | (0.018) | (0.018) | (0.027) | (0.024) | (0.029) | (0.027) |
| Literacy competence | 0.001*** | 0.001*** | 0.001*** | 0.000 | 0.001*** | -0.000 |
| | (0.000) | (0.000) | (0.000) | (0.000) | (0.000) | (0.000) |
| Education in origin country | 0.027** | 0.027** | -0.010 | 0.071*** | -0.010 | 0.075*** |
| | (0.014) | (0.014) | (0.020) | (0.018) | (0.022) | (0.020) |
| Age | -0.002*** | -0.002*** | -0.001 | -0.004*** | -0.001 | -0.005*** |
| | (0.001) | (0.001) | (0.001) | (0.001) | (0.001) | (0.001) |
| Having children | -0.024 | -0.026 | 0.043* | -0.104*** | 0.044* | -0.128*** |
| | (0.017) | (0.017) | (0.023) | (0.024) | (0.024) | (0.027) |
| Leaving with a partner | -0.024 | -0.028* | 0.024 | -0.083*** | 0.055** | -0.074*** |
| | (0.016) | (0.016) | (0.021) | (0.024) | (0.024) | (0.027) |
| Up to 10 years in the country | 0.005 | 0.002 | 0.006 | -0.017 | 0.016 | -0.032 |
| | (0.016) | (0.016) | (0.023) | (0.020) | (0.025) | (0.023) |
| Cultural distance | | | | | -0.041*** | -0.012 |
| | | | | | (0.015) | (0.014) |
| Constant | 0.887*** | 0.729*** | 0.821*** | 0.830*** | 0.958*** | 0.919*** |
| | (0.068) | (0.072) | (0.097) | (0.088) | (0.109) | (0.099) |
| Observations | 4,843 | 4,843 | 2,704 | 2,139 | 2,232 | 1,706 |
| R-squared | 0.083 | 0.091 | 0.092 | 0.068 | 0.107 | 0.079 |

Individuals aged 25–65, all models control for include country fixed effect. Appendix 5 in S1 File presents the same results which include both country of origin and country of destination fixed effects.

Standard errors in parentheses *** p<0.01

** p<0.05

* p<0.1

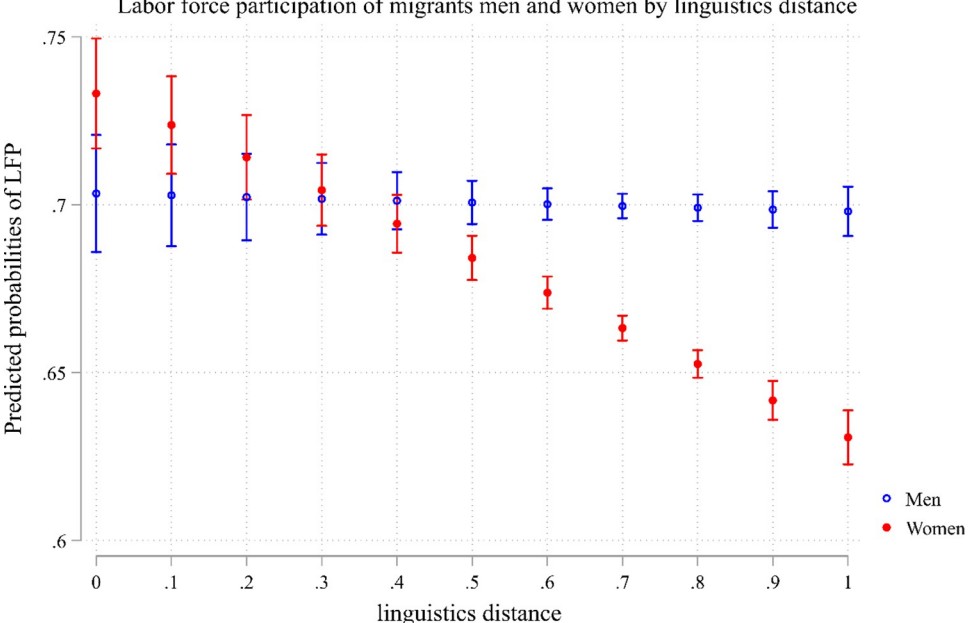

**Fig 1. Labor force participation of migrant men and women by linguistics distance.**

interaction term emerges as negative and statistically significant, indicating that language distance disproportionately affects migrant women while having no discernible impact on migrant men. Moreover, the main effect of gender is now positive and significant, indicating that in the absence of any language distance between their native language and the host country's language, migrant women do not face a significant disadvantage. Fig 1 visually depicts these outcomes based on Model 2, illustrating that while the probability of labor force participation remains unaffected by language distance for migrant men, it decreases for migrant women as language distance increases, thereby widening the gap by gender in terms of labor force participation.

These findings are further substantiated in Model 3 to Model 6, where the sample is disaggregated by gender. Specifically, the influence of language distance on labor market participation for migrant men is slightly positive, whereas for migrant women, it exhibits a substantial, negative, and statistically significant effect, which remains significant even after controlling for cultural distance (Model 5). Appendix 3 in S1 File presents the same result including the beta coefficient and suggests that the magnitudes of the effects of the linguistic distance is twice that of that of linguistic proficiency and cultural distance. The unexpected discovery of a positive correlation between linguistic distance and the labor market participation of migrant men in Model 4 challenges our initial research hypotheses. Various potential explanations emerge from this finding. Firstly, there might be a significant positive selection among male immigrants from countries with greater linguistic disparities. Notably, when controlling for cultural distance in Model 6, the significance of linguistic distance diminishes, lending support to the idea of selection, given that cultural distance is based on place of birth. Additionally, the imperative for men to participate in the labor market to support their families could contribute to this phenomenon. However, it is essential to recognize that labor market participation encompasses both individuals actively seeking employment and those currently employed. Therefore, our subsequent analysis will narrow its focus specifically to employment.

**Table 2. Employment of migrants by linguistics distance.**

| VARIABLES | (1) All | (2) All | (3) Women | (4) Men | (5) Women | (6) Men |
|---|---|---|---|---|---|---|
| Linguistics distance | -0.349*** | -0.151** | -0.539*** | -0.114 | -0.487*** | 0.009 |
| | (0.052) | (0.065) | (0.072) | (0.072) | (0.093) | (0.100) |
| Female | -0.181*** | 0.110* | | | | |
| | (0.014) | (0.058) | | | | |
| Female *Linguistics distance | | -0.399*** | | | | |
| | | (0.077) | | | | |
| BA | 0.080*** | 0.077*** | 0.133*** | -0.014 | 0.132*** | -0.013 |
| | (0.020) | (0.020) | (0.026) | (0.030) | (0.029) | (0.033) |
| MA+ | 0.140*** | 0.139*** | 0.103*** | 0.184*** | 0.119*** | 0.184*** |
| | (0.021) | (0.021) | (0.028) | (0.030) | (0.031) | (0.033) |
| Literacy competence | 0.001*** | 0.001*** | 0.001*** | 0.001*** | 0.001*** | 0.001*** |
| | (0.000) | (0.000) | (0.000) | (0.000) | (0.000) | (0.000) |
| Education in origin country | 0.043*** | 0.043*** | 0.001 | 0.088*** | -0.004 | 0.081*** |
| | (0.015) | (0.015) | (0.021) | (0.022) | (0.023) | (0.025) |
| Age | -0.000 | -0.000 | 0.002 | -0.004*** | 0.001 | -0.005*** |
| | (0.001) | (0.001) | (0.001) | (0.001) | (0.001) | (0.001) |
| Having children | -0.037** | -0.039** | 0.074*** | -0.164*** | 0.068*** | -0.188*** |
| | (0.019) | (0.019) | (0.024) | (0.030) | (0.026) | (0.033) |
| Leaving with a partner | -0.058*** | -0.061*** | 0.015 | -0.147*** | 0.038 | -0.150*** |
| | (0.018) | (0.018) | (0.023) | (0.029) | (0.025) | (0.033) |
| Up to 10 years in the country | -0.007 | -0.010 | -0.011 | -0.037 | -0.003 | -0.039 |
| | (0.018) | (0.018) | (0.025) | (0.025) | (0.027) | (0.029) |
| Cultural distance | | | | | -0.045*** | -0.043** |
| | | | | | (0.016) | (0.017) |
| Constant | 0.685*** | 0.546*** | 0.507*** | 0.785*** | 0.684*** | 0.906*** |
| | (0.076) | (0.081) | (0.103) | (0.109) | (0.116) | (0.122) |
| Observations | 4,843 | 4,843 | 2,704 | 2,139 | 2,232 | 1,706 |
| R-squared | 0.105 | 0.109 | 0.094 | 0.128 | 0.104 | 0.136 |

Individual aged 25–65, all models control for include country fixed effect. Appendix 6 in S1 File presents the same results which include both country of origin and country of destination fixed effects.

Standard errors in parentheses *** p<0.01

** p<0.05

* p<0.1

Table 2 provides an analogous model to Table 1, focusing on actual employment instead of labor force participation. Consistent with the findings in Table 1, language distance exhibits a negative impact on the likelihood of employment, even after accounting for language proficiency (Model 1). Additionally, the interaction term (Model 2) remains significant and negative, indicating the compounded disadvantage experienced by migrant women. However, it is noteworthy that the main effect of language distance is reduced to half of his size once the interaction term is included.

Fig 2 presents a graphical representation of the outcomes derived from Model 2. It demonstrates that, for migrant men, the employment probabilities are just slightly reduced by language distance. However, in the case of migrant women, their employment probabilities decrease as the language distance increases, leading to a widening gender gap in employment

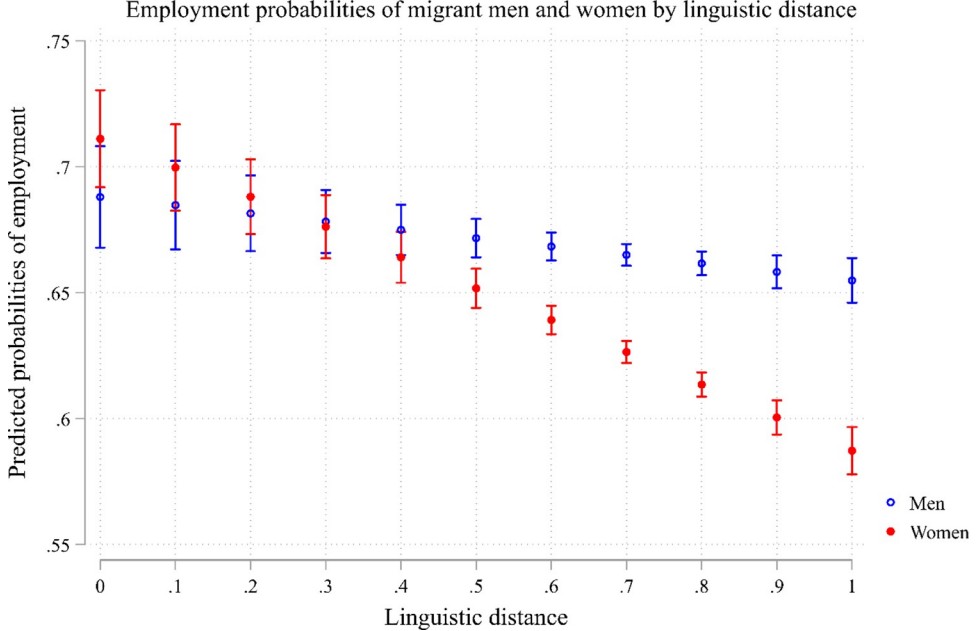

**Fig 2. Employment probabilities of migrant men and women by linguistics distance.**

probabilities. This observation is further reinforced by Model 3 to Model 6, which disaggregate the analysis by gender [3, 4] and control for cultural distance [5, 6], revealing that while language distance has a substantial influence on migrant women, it does not affect migrant men. By considering the disparities between labor force participation and actual employment as indicative of the gap between labor preferences (supply) and employability (demand), we can infer that while language distance influences both aspects of the employment equation for women. For men, language distance might slightly influence the supply side (labor force participation and most probably the active looking for work) while practically not affecting their employment. In this context, language distance affects both the supply side (labor preferences) and the demand side (employability) of employment dynamics for women and less so for men.

We turn now to the effect of language distance on weekly working hours. Table 3 presents the results of linear regression models where the dependent variable is working hours. According to Model 1, migrant women work 6 hours less than migrant men. Language distance reduces working hours by over 5 hours for the maximum distance. Fig 3 presents the results from Model 2 in Table 3, which includes interaction between gender and language distance. On average, migrant women work considerably fewer hours than migrant men, and interestingly, language distance has the same impact on the weekly working hours of migrant men and women, so that a large linguistic distance decreases the working hours by over six hours. Models 3–6 present the same models by gender. While in Models 3 and 4 the effect of linguistics distance is significant for both migrant's men and women, once we control for cultural distance (Models 5 and 6) the effect remains negative and significant just for migrant women. This suggests that for migrant men with the same cultural distance, the linguistics distance has no effect on their working hours, while for migrant women it still reduces their working hours (note that the number of cases is also reduced in Models 5 and 6 compared to 3 and 4).

Lastly, Table 4 presents the results of the linear regression analyses, with ISEI (occupational standing) as the dependent variable. Surprisingly, the impact of language distance is positive

**Table 3. Working hours of migrants by linguistics distance.**

| VARIABLES | (1) All | (2) All | (3) Women | (4) Men | (5) Women | (6) Men |
|---|---|---|---|---|---|---|
| Linguistics distance | -5.475*** | -6.532*** | -5.540** | -5.067** | -7.387** | -3.237 |
| | (1.654) | (1.992) | (2.452) | (2.201) | (3.300) | (3.201) |
| Female | -6.302*** | -7.957*** | | | | |
| | (0.433) | (1.793) | | | | |
| Female *Linguistics distance | | 2.293 | | | | |
| | | (2.412) | | | | |
| BA | -0.722 | -0.717 | 0.070 | -2.263** | -0.389 | -2.252** |
| | (0.620) | (0.620) | (0.850) | (0.899) | (0.968) | (1.018) |
| MA+ | 0.821 | 0.825 | 0.527 | 0.830 | 0.157 | 0.164 |
| | (0.607) | (0.607) | (0.890) | (0.813) | (0.988) | (0.916) |
| Literacy competence | 0.028*** | 0.028*** | 0.017*** | 0.038*** | 0.019*** | 0.043*** |
| | (0.004) | (0.004) | (0.006) | (0.006) | (0.007) | (0.006) |
| Education in origin country | 0.349 | 0.347 | -0.343 | 1.005 | 0.140 | 0.793 |
| | (0.481) | (0.481) | (0.702) | (0.647) | (0.798) | (0.762) |
| Age | 0.056** | 0.057** | 0.108*** | 0.000 | 0.111** | 0.046 |
| | (0.025) | (0.025) | (0.038) | (0.033) | (0.044) | (0.037) |
| Having children | 2.077*** | 2.094*** | 5.110*** | -0.632 | 5.770*** | 0.247 |
| | (0.579) | (0.579) | (0.796) | (0.866) | (0.884) | (0.984) |
| Leaving with a partner | -2.422*** | -2.408*** | -1.316* | -3.241*** | -1.167 | -3.836*** |
| | (0.567) | (0.567) | (0.748) | (0.906) | (0.846) | (1.018) |
| Up to 10 years in the country | 0.639 | 0.664 | -0.268 | 1.044 | -0.143 | 1.373 |
| | (0.576) | (0.577) | (0.868) | (0.764) | (0.992) | (0.893) |
| Cultural distance | | | | | 0.235 | -0.814 |
| | | | | | (0.579) | (0.526) |
| Constant | 31.984*** | 32.720*** | 24.997*** | 32.861*** | 24.611*** | 31.103*** |
| | (2.391) | (2.513) | (3.478) | (3.185) | (3.992) | (3.593) |
| Observations | 3,168 | 3,168 | 1,624 | 1,544 | 1,341 | 1,213 |
| R-squared | 0.132 | 0.132 | 0.071 | 0.140 | 0.070 | 0.152 |

Individuals aged 25–65, all models control for include country fixed effect. Appendix 7 in S1 File presents the same results which include both country of origin and country of destination fixed effects.

Standard errors in parentheses *** p<0.01

** p<0.05

* p<0.1

among migrant women when accounting for factors such as gender, education, language proficiency, and socio-demographic characteristics, while it is insignificant among migrant men. Notably, when examining the sample stratified by gender (Models 3 to 6), the effect of linguistic distance is positive for women and is stronger when we control for cultural distance (Model 5). The finding that linguistic distance shapes the occupational prestige of women suggests that selection into employment might play a role in this aspect. That is, once we control for the decision to participate in the labor market (as these models focus on employed individuals), linguistic distance has a positive effect on the type of occupation in which migrant women are employed, and this is even stronger when controlling for cultural distance. This suggests a strong selection effect. Women who successfully navigate the language barrier to enter the labor market, likely possess greater skills in comparison to their peers, and are probably more

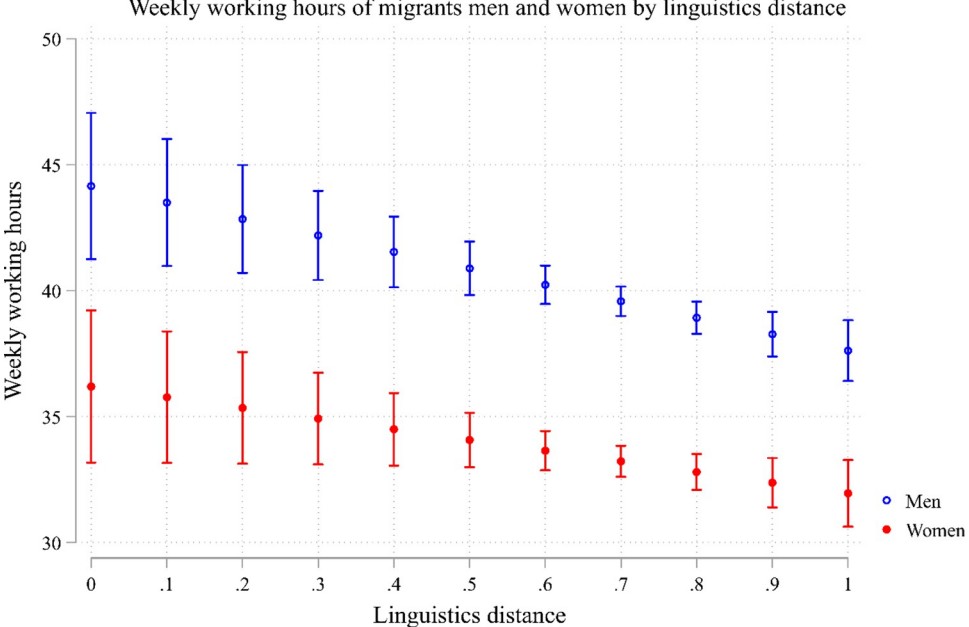

**Fig 3. Weekly working hours of migrant men and women by linguistics distance.**

inclined to pursue lucrative employment opportunities. As the barrier becomes more formidable, the job must increasingly justify the exerted effort.

Appendix 2 in S1 File presents regression models of the association between linguistics distance and the literacy competence of migrants by gender. The purpose of this table is to demonstrate that while the relationship between linguistics distance and various measures of labor market integration differs between men and women, the effect of linguistics distance and language proficiency of both genders does not differ. In other words, linguistics distance is equally significant for language acquisition for both genders, but it has a much greater impact on labor market disadvantage for women. These findings again illustrate how language serves as a more significant barrier for women than men and the marginalization of women in the labor market.

Furthermore, Appendix 4 in S1 File elucidates the impact of literacy proficiency on the labor market outcomes of both migrant men and women. The results reveal that literacy proficiency plays a more significant role in influencing the labor force participation and employment status of migrant women compared to men. Conversely, this pattern is reversed when considering working hours. It is noteworthy that, at least for the entire sample, no gender disparities are discerned in the correlation between literacy proficiency and occupational prestige. We refrain from explicating any potential directionality in the association between literacy proficiency and diverse labor market outcomes. Clearly, this relationship is bidirectional, wherein heightened verbal proficiency correlates with increased engagement in the labor market, while conversely, participation in the labor market is anticipated to enhance linguistic capabilities. However, we underscore that this correlation exhibits greater strength among immigrant women compared to men. Appendix 3 in S1 File presents the results of labor market outcome models, including beta coefficients, categorized by gender to aid in the comparison of coefficients with different scales. It is evident that for migrant women, in most models, the impact of linguistic distance on most labor market outcomes is more significant than cultural distance and literacy competence. However, this is not the case for migrant men. This

**Table 4. Occupational prestige of migrants by linguistics distance.**

| VARIABLES | (1) All | (2) All | (3) Women | (4) Men | (5) Women | (6) Men |
|---|---|---|---|---|---|---|
| Linguistics distance | 0.042 | -4.196* | 4.929* | -4.625 | 14.725*** | 5.204 |
| | (1.999) | (2.384) | (2.798) | (2.854) | (3.533) | (3.777) |
| Female | -5.136*** | -11.766*** | | | | |
| | (0.536) | (2.109) | | | | |
| Female *Linguistics distance | | 9.255*** | | | | |
| | | (2.848) | | | | |
| BA | 8.217*** | 8.234*** | 7.700*** | 9.298*** | 7.024*** | 9.592*** |
| | (0.751) | (0.750) | (0.985) | (1.172) | (1.047) | (1.218) |
| MA+ | 17.930*** | 17.942*** | 20.683*** | 15.830*** | 19.762*** | 15.185*** |
| | (0.767) | (0.766) | (1.106) | (1.070) | (1.152) | (1.109) |
| Literacy competence | 0.086*** | 0.086*** | 0.092*** | 0.080*** | 0.091*** | 0.079*** |
| | (0.005) | (0.005) | (0.008) | (0.007) | (0.008) | (0.008) |
| Education in origin country | -2.401*** | -2.395*** | -3.408*** | -1.407* | -3.356*** | -2.474*** |
| | (0.592) | (0.591) | (0.835) | (0.837) | (0.898) | (0.911) |
| Age | 0.032 | 0.033 | -0.002 | 0.050 | 0.011 | 0.063 |
| | (0.030) | (0.030) | (0.044) | (0.042) | (0.048) | (0.044) |
| Having children | 2.021*** | 2.089*** | 1.898* | 1.646 | 1.921* | 2.880** |
| | (0.729) | (0.728) | (0.970) | (1.133) | (1.021) | (1.175) |
| Leaving with a partner | -3.587*** | -3.553*** | -3.907*** | -2.079* | -4.172*** | -3.042** |
| | (0.720) | (0.718) | (0.890) | (1.224) | (0.947) | (1.275) |
| Up to 10 years in the country | 3.321*** | 3.439*** | 4.409*** | 3.156*** | 4.402*** | 3.210*** |
| | (0.701) | (0.701) | (1.019) | (0.974) | (1.093) | (1.041) |
| Cultural distance | | | | | -1.710*** | -3.009*** |
| | | | | | (0.632) | (0.620) |
| Constant | 14.972*** | 17.974*** | 6.823* | 17.835*** | 4.992 | 20.594*** |
| | (2.867) | (3.007) | (3.980) | (4.052) | (4.290) | (4.237) |
| Observations | 2,626 | 2,626 | 1,347 | 1,279 | 1,229 | 1,136 |
| R-squared | 0.400 | 0.402 | 0.464 | 0.343 | 0.461 | 0.372 |

Individual aged 25–65, all models control for include country fixed effect. Appendix 8 in S1 File presents the same results which include both country of origin and country of destination fixed effects.

Standard errors in parentheses *** p<0.01

** p<0.05

* p<0.1

suggests that linguistic distance captures a crucial concept that shapes the integration of migrant women.

## 6. Conclusion

This study aimed to investigate how linguistic distance shapes migrants' labor market status, focusing on gender differences. Specifically, we examine how linguistic distance, independent of literacy skills and cultural distance, influences migrants' labor force participation, working hours, and occupational prestige of migrant men and women. Our findings indicate that linguistics distance shapes labor market outcomes net of language skills, cultural distance, and education, mainly for women. Thus, we claim that linguistic distance serves as a proxy for additional cultural aspects that are not grasped by source and destination cultural distance

measured by Inglehart et al. (2014) [50], and hence is related to labor market integration not due to merits but due to social distance. The gender aspect of the effect of language distance is essential. In line with previous studies [52], we show that migrant women from countries more linguistically remote from their destination are less prone to take part in the labor market and be employed. By controlling for language ability and education, we can identify that the roots of migrant women's disadvantage are probably social and cultural rather than human capital.

One important question that our findings raise is the mechanism through which language distance affects labor market integration. Scholars of cultural capital would perceive language distance as a form of cultural capital. The inability "to pass" as native (or as coming from a similar origin to natives) serves as a basis for labor market discrimination [27]. Women, who are more likely to work in occupations that require communication skills [30, 31], are more vulnerable to such discrimination. On the other hand, scholars coming from the gender-cultural norms approach view language distance as a measurement of cultural traits that are important to the labor market. Such scholars primarily identify home-work preferences as a cultural trait that is captured by language distance [12, 15]. Hence, language distance is expected to have a stronger effect on women than on men. In essence, the difference between these approaches is in the agency: while the former scholars put more emphasis on labor market discrimination and the ability of employers to prefer native language speakers over other employees, the latter put more emphasis on the agency of the immigrants and their cultural preferences related to the gender division of work.

Our results support the gender cultural norms perspective to a large degree. We found that the impact of language distance is evident in labor force participation, employment, and working hours of migrant women, which supports the gender cultural norms perspective. Entry into the labor market and weekly working hours are usually regarded as a result of preference rather than discrimination. At the same time, the effect on occupational status is positive, so that larger linguistic distance is associated with higher occupational prestige of migrant women. Under the gender-cultural norms approach, we would expect not to see negative effect on occupational prestige as the selection process in entry to the labor market would result in a positive effect of language distance on occupational scores (since only the most skillful migrant women would enter the labor market, their gains would be higher when there is no discrimination against them). The findings provide only a weak support for the cultural capital perspective, as evidenced by the lack of effect on migrant women's working hours and the persistence of the results even after controlling for cultural distance. However, the unintuitive results regarding occupational standing suggest that more research is needed in order to understand the mechanism of the performative aspect of language distance. Hence, a study of migrants' assimilation within specific occupations is much needed.

Our findings suggest that language distance is an important factor for both men and women in their ability to acquire the destination language (see Appendix 2 in S1 File). However, the impact of language distance on labor market integration is much greater for women than men. This means that women are more likely to experience labor market disadvantages if they have a considerable language distance, regardless of their proficiency in the language used in their destination. These results suggest that migrant women are more likely to face additional barriers in the labor market. It is plausible that decisions regarding the division of work within the family play a significant role in shaping the labor market outcomes of migrant women, particularly in terms of their participation and employment. Nonetheless, discrimination and bias related to cultural distance might also exacerbate the impact of language distance on migrant women's career prospects.

Overall, these findings highlight that while policies and programs that support language acquisition might improve the language abilities of migrant men and women, they may not effectively combat the gendered barriers women face in the labor market. By promoting equal opportunity and addressing issues and cultural norms related to the division of work and care within the family, we can help create a more equitable and inclusive labor market for migrant men and women. It is important to address both linguistic and gendered barriers to ensure that all individuals have an equal chance to succeed in the labor market.

## Supporting information

**S1 File.**
(DOCX)

## Author Contributions

**Conceptualization:** Eyal Bar-Haim, Debora Pricila Birgier.

**Data curation:** Eyal Bar-Haim, Debora Pricila Birgier.

**Formal analysis:** Eyal Bar-Haim, Debora Pricila Birgier.

**Funding acquisition:** Debora Pricila Birgier.

**Investigation:** Eyal Bar-Haim, Debora Pricila Birgier.

**Methodology:** Eyal Bar-Haim, Debora Pricila Birgier.

**Project administration:** Eyal Bar-Haim, Debora Pricila Birgier.

**Validation:** Eyal Bar-Haim, Debora Pricila Birgier.

**Visualization:** Eyal Bar-Haim, Debora Pricila Birgier.

**Writing – original draft:** Eyal Bar-Haim, Debora Pricila Birgier.

**Writing – review & editing:** Eyal Bar-Haim, Debora Pricila Birgier.

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
