## [Decision Letter · Decision Letter 0]

14 Nov 2023

PONE-D-23-27134Language Distance and Labor Market Integration of Migrants: Gendered PerspectivePLOS ONE

Dear Dr. Birgier,

Thank you for submitting your manuscript to PLOS ONE. After careful consideration, we feel that it has merit but does not fully meet PLOS ONE’s publication criteria as it currently stands. Therefore, we invite you to submit a revised version of the manuscript that addresses the points raised during the review process.

I find the exploration of factors influencing immigrant labor market integration intriguing. While the results presented are captivating, the reviewers and I do have some concerns regarding their interpretation. Specifically, I would like to suggest the following areas for enhancement (as well as the comments from the reviwers):

Given your assertion that "linguistic distance" encapsulates "cultural distance," it might be beneficial to incorporate more direct metrics of "cultural distance" in the analysis.

As the current analysis only delves into gender disparities concerning the impact of "cultural distance," expanding the study to encompass potential gender-specific effects of "linguistic proficiency," as discussed in the literature review but not explored empirically, could add depth to the research.

Exploring or, at the very least, acknowledging additional potential limitations of the analysis, such as the likelihood of endogeneity and reverse causality linked to the linguistic proficiency measure derived from PIAAC, would be advantageous.

We look forward to receiving your revised manuscript.

Kind regards,

Jolanta Maj

Academic Editor

PLOS ONE

“We would also like to acknowledge the Israel Science Foundation (80/20) and the Swedish Research Council for Health, Working Life, and Welfare (FORTE) (2016-07105) for their fnancial support of this work.”

Reviewers' comments:

Reviewer's Responses to Questions

**Comments to the Author**

1. Is the manuscript technically sound, and do the data support the conclusions?

Reviewer #1: Yes

Reviewer #2: Yes

2. Has the statistical analysis been performed appropriately and rigorously? 

Reviewer #1: Yes

Reviewer #2: Yes

3. Have the authors made all data underlying the findings in their manuscript fully available?

Reviewer #1: Yes

Reviewer #2: Yes

4. Is the manuscript presented in an intelligible fashion and written in standard English?

Reviewer #1: Yes

Reviewer #2: Yes

5. Review Comments to the Author

Reviewer #1: Review on “Language Distance and Labor Market Integration of Migrants: Gendered Perspective”

This paper investigates the gendered effect of language distance on immigrant labor market integration. The author’s hypothesis is that linguistic distance is a proxy of cultural distance, and therefore influences immigrant integration over and above proficiency in the destination language. The author uses PIAAC (OECD) data on several countries and linear probability and linear regression models. The author’s main result is that linguistic distance affects various measures of women’s labor market integration but not of men’s.

I think that the paper addresses an important topic, the analysis is generally competently done. Yet, I think that there is significant room for improvement. In what follows, I provide some suggestions.

Main comments

1) Hypotheses. Reading the hypotheses, the differences between the “cultural capital” and the “cultural distance” hypotheses are not very clear to me. Both seem to be “supply driven” explanations (i.e. from the side of the worker) and none of them is a “demand driven” explanation (e.g. employer’s discrimination) of the higher/lower labor market integration of migrants. As such, I cannot really appreciate the difference between the two. Moreover, while in my opinion “cultural capital” seems to be one-sided that is it should characterize one culture irrespective of the destination culture, the cultural distance explanation seems to be two-sided (dyadic), so one culture (i.e. migrants from a given origin country) may perform differently in different destination countries. Not having fully understood the differences (perhaps because I am less familiar with the sociological literature), it is really hard to me (and potentially also for the average reader) to evaluate the tests that the authors provide in the empirical section. Following the above line of reasoning, for instance, one could hypothesize that immigrants from a given culture should perform very similarly in all destination countries according to the cultural capital explanation (i.e. no differences across destination countries of immigrants from a given culture, for instance from a traditional culture that posits that women should not work), while according to the second explanation (cultural distance), for a given origin culture there should be differences across destination countries depending on the distance between the two cultures. So, the significance of the “linguistic distance” variable in the regression should support the “cultural distance” explanation only. Moreover, I cannot always follow the arguments of the author. For instance, in section 3 the author mentions the use of the origin language at home as a form of commitment towards one’s own culture, but then provides hypotheses formulated in terms of “linguistic distance” without any reference to the use of the origin language at home. So, it is not always easy to follow the arguments of the author.

2) “Linguistic distance” as a proxy of “cultural distance”. The core of the author’s argument is that linguistic distance has an effect over and above linguistic proficiency in the destination language because it captures immigrants’ cultural traits. However, a corollary would be that with a good measure of “cultural distance” included in the regressions, linguistic distance should not be significant in explaining immigrant labor market integration. Thus, I suggest the author to try and use other measures of distance that have been employed in the literature (e.g. genetic distance, or potentially even measures of “cultural distance” if available, perhaps built using the World Values Survey or similar surveys).

3) Gender differences in the effect of linguistic proficiency. I think that the author should devote more space to the potential gendered effects of linguistic proficiency and highlight the results of such an analysis especially if it has not been carried out on cross-country data (but mainly with single-country data). I would like to see the results of a regression including not only the interaction between linguistic distance and gender, but also between linguistic proficiency and gender. Moreover, to compare the magnitudes of the effects of the two variables (linguistic proficiency and linguistic distance), it would be useful to standardize them (so as they have mean zero and unit standard deviation). This way, the coefficient could be interpreted as the effect on the dependent variables of increasing the independent variables by one standard deviation.

4) Origin country FEs. To the best of my understanding, the author includes in the model destination country, but not origin country fixed effects. In the model without gender interactions, this could be motivated by the high correlation between indicators of origin countries and linguistic distance (unless there are several mother tongues observed with one country). However, omitting the origin country FEs introduces in the analysis a potential confounder, i.e., linguistic distance may capture discrimination not based on culture (e.g. racial discrimination). However, in the model in which gender*linguistic distance interactions are included, destination country FEs could be included. For instance, in a model with both gender*linguistic distance, and gender*linguistic proficiency interactions, I would be interested in observing which coefficients remain significant in the regression after including both origin and destination countries fixed effects.

5) Endogeneity of linguistic proficiency. There is a rich literature in economics that aims to tackle the potential endogeneity of linguistic proficiency. As for PIAAC literacy scores, there is even a potential reverse-causality issue, because the PIAAC literature suggests that participation in the labor market (or job-related variables) may affect such scores. This happens certainly for numeracy, so the authors should look for similar evidence for verbal skills. The authors should discuss how their paper is positioned in the literature. At present, the issue is neither discussed nor addressed in the paper.

6) Strange results. Some results are hard to explain. For instance, the positive effect of linguistic distance on men’s LFP in Table 1. One could think of a household labor supply model. As both partners (spouses) are likely to have the same linguistic distance, if they are together in the same destination country and speak the same mother tongue, it might be the case that if linguistic distance reduces female’s labor force participation, then men are more likely to be in the labor force. If might be interesting to interact marital status, or an indicator for the spouse being in the destination country vs. having remained at home with linguistic proficiency and linguistic distance indicators in columns 3 and 4 of Table 1.

7) Working of the hypotheses in the “real world”. While speaking well/poorly a language is something that can be easily observed by the employer, and the effect is likely to be “demand driven” (e.g. workers not speaking well Italian are not hired in some jobs, or are hired in manual jobs or jobs in which communication skills are not necessary), the working of the “cultural distance” and “cultural capital” hypotheses is not clear. For instance, at p. 6 the author writes for the “cultural capital”: “The ability to pass as a native, or to come from a similar background as locals, becomes the basis or discrimination in the labor market”. I wonder how can the employer observe “culture”? S/he can probably observe ethnicity and then infer culture. The employer for sure does not observe linguistic distance. So again, how the model performs including controls for ethnicity or ethnic backgrounds (even aggregated, Middle East, South Asia, China, Pakistan-India, etc.). Incidentally, this definition of “cultural capital” effects contradicts my interpretation above of the “cultural capital” as being a supply-driven explanation (and being a two-sided explanation), but it sounds like a demand-driven explanation (since the author calls upon discrimination). So once again a clearer description of the main differences between the hypotheses is needed.

The paper as I would do it

(this should not be necessarily followed in the revision process, it reflects a different way of tackling the problem).

I think that the cultural distance explanation could remain in the paper, but it would be better to use a more direct measure (see above). I would start with a model using linguistic proficiency and cultural distance, both interacted with gender, and comment the results. Then, based on the empirical literature that used linguistic distance as an instrument (instrumental variables) to estimate the causal effect of linguistic distance, I would say that the regressions could be affected by endogeneity bias and estimate the model with linguistic distance and cultural distance both interacted by gender (omitting linguistic proficiency). The linguistic distance (and the interaction with gender) are assumed to capture the effect of linguistic proficiency, or at least the presumably exogenous variation deriving from linguistic distance. The cultural distance term will directly capture cultural distance. The model would be a reduced-form model in the Instrumental variables literature.

Reviewer #2: Dear Authors,

I was really interested in reviewing your article. Sorry, for the delay, I've been sick and still am a bit.

However, I have some recommendations for how your article could be improved a bit:

1. The concept of intersectionality, as mentioned, is indeed broader than just gender and language. It encompasses a wide range of factors such as race, age, and name, which interact and intersect to create unique experiences and challenges for individuals. Please explore some examples of how intersectionality plays out in the context of labour market integration to make it catchier for the reader. In the current text, you mention it, and this is it. Intersectionality can be observed when migrant women face both gender-related discrimination and language barriers. For instance, women from certain cultural backgrounds may encounter gender-specific biases that intersect with language differences, making their journey into the labour market even more challenging.

2. Also, intersectionality is not limited to migrant women. Native women can also experience it, as factors like race, age, and qualifications intersect to shape their employment opportunities and experiences. For instance, an older native woman from a minority background may face unique challenges in the labour market compared to her younger counterparts.

3. Labour market challenges are evident in the lower qualification levels of women from specific countries of origin. These women may face language barriers and encounter differences in cultural attitudes towards work and employability, which can further affect their integration into the labour market.

4. You would not that often find highly qualified women who show such "big" challenges. Often the qualification is not recognised, and they are victims of deskilling, but mainly, they find a way to the labour market. That means on the contrary, that you find this phenomenon with women who did not work in countries of origin, or at least not in academic jobs etc.

5. Following this, it means that language is not the only explanation here, but a further hint for a complex phenomenon.

6. This also highlights that women from certain cultures may not have the right (by culture, by family, by standing, by capital) to participate fully in the culture of the receiving country. The example of the guest worker generation in Germany illustrates how cultural differences and language barriers can persist over generations, affecting labour market participation.

7. Both men and women can be influenced by their working environment regarding language requirements. Some workplaces may demand minimal language skills, while others require more extensive proficiency. This intersection of language skills and the work environment can significantly impact individuals' job prospects.

8. The article emphasises the crucial role of language in shaping an individual's reality and participation in the host society. When migrants lack proficiency in the language of the receiving country, they may find it challenging to integrate into the working culture fully. This can lead to the employment of migrants by co-ethnic employers who share the same language and cultural background.

See, if here and there, you can bring in more examples or even consider the fact that language constitutes the worldview. We know from studies, that for instance, people who are bilingual, have another (maybe richer world conceptualisation, than monolingual people). If you can deepen this a bit before you start with the data, would make more sense for the reader.

Kind regards

6. PLOS authors have the option to publish the peer review history of their article (what does this mean?). If published, this will include your full peer review and any attached files.

Reviewer #1: No

Reviewer #2: **Yes: **Alexandra David

---

## [Author Response · Author response to Decision Letter 0]

17 Jan 2024

Dear Prof. Jolanta Maj,

We are delighted to submit a thoroughly revised version of our manuscript titled “Language Distance and Labor Market Integration of Migrants: Gendered Perspective” (PONE-D-23-27134) for your consideration. We appreciate the opportunity to revise the paper and would like to express our sincere gratitude to you and both reviewers for their invaluable advice, as well as detailed and constructive comments.

Both Reviewer 1 and Reviewer 2 provided insightful comments that pinpointed specific ways to enhance the manuscript. We found all of their suggestions to be constructive and have diligently addressed each of them, as outlined in our comprehensive response letter.

Reviewer #1

1) Hypotheses. Reading the hypotheses, the differences between the "cultural capital" and the "cultural distance" hypotheses are not very clear to me. Both seem to be "supply-driven" explanations (i.e. from the side of the worker) and none of them is a "demand-driven" explanation (e.g. employer's discrimination) of the higher/lower labor market integration of migrants. As such, I cannot really appreciate the difference between the two. Moreover, while in my opinion "cultural capital" seems to be one-sided that is it should characterize one culture irrespective of the destination culture, the cultural distance explanation seems to be two-sided (dyadic), so one culture (i.e. migrants from a given origin country) may perform differently in different destination countries. Not having fully understood the differences (perhaps because I am less familiar with the sociological literature), it is really hard for me (and potentially also for the average reader) to evaluate the tests that the authors provide in the empirical section.

Following the above line of reasoning, for instance, one could hypothesize that immigrants from a given culture should perform very similarly in all destination countries according to the cultural capital explanation (i.e. no differences across destination countries of immigrants from a given culture, for instance from a traditional culture that posits that women should not work), while according to the second explanation (cultural distance), for a given origin culture there should be differences across destination countries depending on the distance between the two cultures.

So, the significance of the “linguistic distance” variable in the regression should support the “cultural distance” explanation only. Moreover, I cannot always follow the arguments of the author. For instance, in section 3 the author mentions the use of the origin language at home as a form of commitment towards one’s own culture, but then provides hypotheses formulated in terms of “linguistic distance” without any reference to the use of the origin language at home. So, it is not always easy to follow the arguments of the author.

- We rewrote the hypotheses to fully represent this explanation. In addition, we changed the terms used to be (1) cultural capital and (2) gender cultural norms. We agree with the reviewer that the definition was confusing so in the revised manuscript we used the new terms which we hope are clearer and more distinct. 

- We added clarification regarding the theoretical approach represented in the term "cultural capital". Cultural capital, dating back to Bourdieu (2011) is a signaling characteristic – it needs a receptor (teacher, employer, etc.). The student or the employee exhibits cultural capital which the receptor interprets as similar or not similar to the dominant culture. As such, its effect lies in the interaction between the supply and demand. 

- We employ the primary language an individual studied, and the PIAAC query is phrased as follows: “What is the language that you first learned at home in childhood and still understand?” We did not include the languages spoken at home, as this variable has a smaller number of cases due to many individuals discontinuing the use of their mother tongue at home as they grow older. Furthermore, we elucidate the rationale for utilizing this variable in the text as follows: “Therefore, we use language distance as a proxy for cultural norms, including for gender norms. In that line of argument, controlling for other aspects of cultural distance enables us to assess the distinct effect of linguistic distance. While our focus isn't on home language, we believe examining first language acquisition at home might capture childhood exposure to gender norms.” See page 8. 

2) "Linguistic distance" as a proxy of "cultural distance". The core of the author's argument is that linguistic distance has an effect over and above linguistic proficiency in the destination language because it captures immigrants' cultural traits. However, a corollary would be that with a good measure of "cultural distance" included in the regressions, linguistic distance should not be significant in explaining immigrant labor market integration. Thus, I suggest the author try and use other measures of distance that have been employed in the literature (e.g. genetic distance, or potentially even measures of "cultural distance" if available, perhaps built using the World Values Survey or similar surveys).

- We thank the reviewer for this important suggestion. We now included in our model a measurement of “cultural distance” from the world value survey (Inglehart and Welzel (2005)). Contrary to the reviewer and our expectations, adding controls for cultural distance to our Models (numbers 5-6 in each table) does not fully diminish the effect of linguistics distance. This is probably because these two measurements of culture are far from being in full correlation with each other (see page 11 in the paper). 

3) Gender differences in the effect of linguistic proficiency. I think that the author should devote more space to the potential gendered effects of linguistic proficiency and highlight the results of such an analysis especially if it has not been carried out on cross-country data (but mainly with single-country data). I would like to see the results of a regression including not only the interaction between linguistic distance and gender, but also between linguistic proficiency and gender. Moreover, to compare the magnitudes of the effects of the two variables (linguistic proficiency and linguistic distance), it would be useful to standardize them (so that they have mean zero and unit standard deviation). This way, the coefficient could be interpreted as the effect on the dependent variables of increasing the independent variables by one standard deviation.

- This aspect, although not initially the focal point of our study, has gained significance with the insightful input from the reviewer. The analysis prompted by the reviewer's suggestion reveals that the impact of linguistic proficiency on various labor market outcomes varies across genders. Notably, linguistic proficiency plays a more crucial role in the labor force participation, employment, and occupational prestige of migrant women. Conversely, when considering working hours, the influence of linguistic proficiency takes on an opposite effect. Refer to Appendix 4 for the detailed results of these models, and we discuss them on page 16. 

- We ran the models suggested by the reviewer and presented the betta coefficients (see Appendix 3). As can be seen, the results suggest that for migrant women the effect of linguistic distance is not less important than that of cultural distance and of linguistic proficiency for all dimensions except that of occupational prestige. 

4) Origin country FEs. To the best of my understanding, the author includes in the model destination country, but not origin country fixed effects. In the model without gender interactions, this could be motivated by the high correlation between indicators of origin countries and linguistic distance (unless there are several mother tongues observed with one country). However, omitting the origin country FEs introduces in the analysis a potential confounder, i.e., linguistic distance may capture discrimination not based on culture (e.g. racial discrimination). However, in the model in which gender*linguistic distance interactions are included, destination country FEs could be included. For instance, in a model with both gender*linguistic distance, and gender*linguistic proficiency interactions, I would be interested in observing which coefficients remain significant in the regression after including both origin and destination countries fixed effects.

- The reviewer's observation is valid; we initially omitted the inclusion of country-of-origin fixed effects, focusing solely on destination fixed effects. In response, we have incorporated appendices tables 5-8, which replicate the models presented in the main paper while also incorporating country-of-origin fixed effects. Fortunately, most of the results from these models align with our primary arguments in the paper. Notably, the significance of linguistic distance remains a key factor in the labor market integration of migrant women.

5) Endogeneity of linguistic proficiency. There is a rich literature in economics that aims to tackle the potential endogeneity of linguistic proficiency. As for PIAAC literacy scores, there is even a potential reverse-causality issue, because the PIAAC literature suggests that participation in the labor market (or job-related variables) may affect such scores. This happens certainly for numeracy, so the authors should look for similar evidence for verbal skills. The authors should discuss how their paper is positioned in the literature. At present, the issue is neither discussed nor addressed in the paper.

- We added the following sentence to the Data and Sample section: “It is imperative to acknowledge that the assessment of linguistic proficiency derived from PIAAC is contingent upon the language of the destination. Consequently, the consideration of endogeneity issues becomes pivotal, given the sample's constraint to individuals possessing the requisite proficiency to undertake the evaluation (e.g., those with sufficient linguistic competence to comprehend the posed questions)”. We also refer to the paper by Maehler, Martin, and Rammstedt, (2017). In addition, we added in the result section on page 16 a reflection on the bidirectional of association between literacy proficiency and labor market outcomes.

6) Strange results. Some results are hard to explain. For instance, the positive effect of linguistic distance on men's LFP in Table 1. One could think of a household labor supply model. As both partners (spouses) are likely to have the same linguistic distance, if they are together in the same destination country and speak the same mother tongue, it might be the case that if linguistic distance reduces female's labor force participation, then men are more likely to be in the labor force. It might be interesting to interact marital status, or an indicator for the spouse being in the destination country vs. having remained at home with linguistic proficiency and linguistic distance indicators in columns 3 and 4 of Table 1.

- We acknowledge the assertion that this result is indeed unconventional, and as a response, we have addressed it directly in the text (refer to page 13). The distinction highlighted by the reviewer forms the foundation of our discussion on that specific page. In the present findings, it becomes evident that, upon controlling for cultural distance (derived from the individual's country of birth), there is no discernible positive effect of linguistic distance in the case of men. Furthermore, we conducted the analysis suggested by the reviewer, and upon incorporating the interaction into the model, the significance of the effect of linguistic distance on labor force participation diminishes among men.

7) Working on the hypotheses in the "real world". While speaking well/poorly a language is something that can be easily observed by the employer, and the effect is likely to be "demand-driven" (e.g. workers not speaking well Italian are not hired in some jobs, or are hired in manual jobs or jobs in which communication skills are not necessary), the working of the "cultural distance" and "cultural capital" hypotheses is not clear. For instance, at p. 6 the author writes for the "cultural capital": "The ability to pass as a native, or to come from a similar background as locals, becomes the basis or discrimination in the labor market". I wonder how can the employer observe "culture"? S/he can probably observe ethnicity and then infer culture. The employer for sure does not observe linguistic distance. So again, how the model performs including controls for ethnicity or ethnic backgrounds (even aggregated, Middle East, South Asia, China, Pakistan-India, etc.). Incidentally, this definition of “cultural capital” effects contradicts my interpretation above of the “cultural capital” as being a supply-driven explanation (and being a two-sided explanation), but it sounds like a demand-driven explanation (since the author calls upon discrimination). So once again a clearer description of the main differences between the hypotheses is needed.

- As previously mentioned, we have revised the terminology used in the paper to refer to (1) cultural capital and (2) gender cultural norms. The concept of "gender cultural norms" aligns with the rationale of cultural perceptions regarding women's roles and is linked to the division of work and care within families. Its impact is "supply-driven," influencing the motivation to seek employment or increase working hours. On the other hand, "cultural capital," as previously elucidated, involves an interaction between the supply and demand sides, with employers exhibiting a reduced inclination to hire individuals with low cultural capital. In this context, language distance serves as an indicator of low cultural capital. In response to the reviewer's feedback, we have reworded the hypotheses section and made revisions to certain portions of Section 2.2.

- Additionally, concerning the impact of source country variation, we have included an appendix featuring country of origin fixed effects. The analysis reveals that the significance of the linguistic distance effect persists (see Appendix 5-8)

Reviewer 2:

1) The concept of intersectionality, as mentioned, is indeed broader than just gender and language. It encompasses a wide range of factors such as race, age, and name, which interact and intersect to create unique experiences and challenges for individuals. Please explore some examples of how intersectionality plays out in the context of labour market integration to make it catchier for the reader. In the current text, you mention it, and this is it. Intersectionality can be observed when migrant women face both gender-related discrimination and language barriers. For instance, women from certain cultural backgrounds may encounter gender-specific biases that intersect with language differences, making their journey into the labour market even more challenging.

- Thank you for your insightful feedback on the concept of intersectionality in the context of labor market integration. We appreciate your suggestion to delve deeper into examples that illustrate how intersectionality plays out in this specific context. Here is the text that we added to the paper on page 5 “Intersectionality, as a concept extending beyond gender and migration, serves as a crucial lens for understanding the intricate web of challenges individuals face in the labor market. While intersectionality is not restricted to migrant women but rather unfolds when factors like race, age, and qualifications intersect to shape the experience of individuals, in this paper we focus on the interaction between migration and gender and how it might be shaped by linguistic distance.”

2) Also, intersectionality is not limited to migrant women. Native women can also experience it, as factors like race, age, and qualifications intersect to shape their employment opportunities and experiences. For instance, an older native woman from a minority background may face unique challenges in the labour market compared to her younger counterparts.

- Thank you for highlighting the importance of acknowledging that intersectionality extends to native women as well. We fully agree with this perspective, and we have incorporated it into the text to ensure a more comprehensive exploration of how factors like race, age, and qualifications intersect to shape employment opportunities and experiences for both migrant and native women (see page 5)

3) Labour market challenges are evident in the lower qualification levels of women from specific countries of origin. These women may face language barriers and encounter differences in cultural attitudes towards work and employability, which can further affect their integration into the labour market.

- Certainly, we appreciate your acknowledgment of the challenges associated with different qualification levels. While we agree with the notion, it's important to mention that due to the size of our sample, a detailed exploration and division of the sample based on varied qualifications was not feasible. As a result, our analysis focused on controlling for the levels of education to capture a broader understanding of the labor market challenges faced by migrant women due to linguistic distance. In addition, in the new revision of the paper, we also added appendices 5-8 which control for the country-of-origin FE which goes in line with your suggestion of the effect of linguistics distance for women from specific countries. We show that even after controlling for that the effect of linguistics distance for migrant women is significant.

4) You would not that often find highly qualified women who show such "big" challenges. Often the qualification is not recognized, and they are victims of deskilling, but mainly, they find a way to the labour market. That means on the contrary, that you find this phenomenon with women who did not work in countries of origin, or at least not in academic jobs etc.

- Thank you for this important comment. It helped us to better understand the differences in the result for ISEI from other labor market characteristics - Women who can get over the language barrier in entering the labor market, which are probably more equipped to do it among their peers, would do it for a lucrative job. When the barrier is higher, the job needs to be more and more worth the effort.

5) Following this, it means that language is not the only explanation here, but a further hint for a complex phenomenon. This also highlights that women from certain cultures may not have the right (by culture, by family, by standing, by capital) to participate fully in the culture of the receiving country. The example of the guest worker generation in Germany illustrates how cultural differences and language barriers can persist over generations, affecting labour market participation.

- We concur with your observation, and to some extent that is exactly our main argument in this paper. We've incorporated this perspective into the text, emphasizing that the challenges faced by women in the context of labor market integration are not solely attributed to language barriers, as we show that even when controlling for language abilities they are disadvantaged in the labor market. This acknowledgment indicates that various factors contribute to the observed challenges beyond language alone.

- Not that the funders had no role in study design, data collection and analysis, decision to publish, or preparation of the manuscript.

We extend our sincere appreciation for their conscientious and constructive feedback, which has significantly contributed to the improvement of our paper. Once again, we thank the Editor and both Reviewers for their valuable contributions.

Sincerely,

Debora P. Birgier and Eyal Bar-Haim

---

## [Decision Letter · Decision Letter 1]

19 Feb 2024

Language Distance and Labor Market Integration of Migrants: Gendered Perspective

PONE-D-23-27134R1

Dear Dr. Birgier

We’re pleased to inform you that your manuscript has been judged scientifically suitable for publication and will be formally accepted for publication once it meets all outstanding technical requirements.

Kind regards,

Jolanta Maj

Academic Editor

PLOS ONE

Additional Editor Comments (optional):

Congratulations on successfully incorporating all the remarks from the reviewers into your paper, "Language Distance and Labor Market Integration of Migrants: Gendered Perspective." Your diligence in addressing the feedback is commendable. The revisions have undoubtedly strengthened the quality of your work. Best wishes for the continued success of your research.

Reviewers' comments:

Reviewer's Responses to Questions

**Comments to the Author**

1. If the authors have adequately addressed your comments raised in a previous round of review and you feel that this manuscript is now acceptable for publication, you may indicate that here to bypass the “Comments to the Author” section, enter your conflict of interest statement in the “Confidential to Editor” section, and submit your "Accept" recommendation.

Reviewer #1: All comments have been addressed

Reviewer #2: All comments have been addressed

2. Is the manuscript technically sound, and do the data support the conclusions?

Reviewer #1: Yes

Reviewer #2: Yes

3. Has the statistical analysis been performed appropriately and rigorously? 

Reviewer #1: Yes

Reviewer #2: Yes

4. Have the authors made all data underlying the findings in their manuscript fully available?

Reviewer #1: No

Reviewer #2: Yes

5. Is the manuscript presented in an intelligible fashion and written in standard English?

Reviewer #1: Yes

Reviewer #2: Yes

6. Review Comments to the Author

Reviewer #1: I am happy with the revisions made by the authors. There might be some small typos:

"hens," do they mean hence?

"labour outcome", I would prefer the plural, "labour outcomes."

Reviewer #2: Thank you one again for giving me the possibility for reviewing your interesting paper.

Good luck for the future

7. PLOS authors have the option to publish the peer review history of their article (what does this mean?). If published, this will include your full peer review and any attached files.

Reviewer #1: No

Reviewer #2: **Yes: **Alexandra David

---

## [Editor Report · Acceptance letter]

7 Mar 2024

PONE-D-23-27134R1 

PLOS ONE

Dear Dr. Birgier, 

I'm pleased to inform you that your manuscript has been deemed suitable for publication in PLOS ONE. Congratulations! Your manuscript is now being handed over to our production team.

Kind regards, 

on behalf of

Dr. Jolanta Maj 

Academic Editor

PLOS ONE